# Aquaphotomics Reveals Subtle Differences between Natural Mineral, Processed and Aged Water Using Temperature Perturbation Near-Infrared Spectroscopy

Yasuhiro Kato [1,2,*,†] , Jelena Munćan [3,†] , Roumiana Tsenkova [3], Dušan Kojić [4], Masato Yasui [4,5], Jing-Yu Fan [1,2] and Jing-Yan Han [1,2,*]

1    Department of Integration of Chinese and Western Medicine, School of Basic Medical Science, Peking University, Beijing 100191, China; jingyuf@yahoo.com
2    Tasly Microcirculation Research Center, Peking University Health Science Center, Beijing 100191, China
3    Aquaphotomics Research Department, Faculty of Agriculture, Kobe University, Kobe 658-8501, Japan; jmuncan@people.kobe-u.ac.jp (J.M.); rtsen@kobe-u.ac.jp (R.T.)
4    Department of Pharmacology, School of Medicine, Tokyo 160-8582, Japan; dusan.s.kojic@gmail.com (D.K.); myasui@a3.keio.jp (M.Y.)
5    Keio Global Research Institute, Keio University, Tokyo 108-8345, Japan
*    Correspondence: katoyasu@z3.keio.jp (Y.K.); hanjingyan@bjmu.edu.cn (J.-Y.H.); Tel.: +81-3-5363-3750 (Y.K.); +86-10-8280-2862 (J.-Y.H.)
†    Both authors contributed to writing of the manuscript equally.

**Abstract:** Current approaches to the quality control of water are unsatisfying due to either a high cost or the inability to capture all of the relevant information. In this study, near-infrared spectroscopy (NIRS) with aquaphotomics as a novel approach was assessed for the discrimination of natural, processed and aged water samples. Temperature perturbation of water samples was employed to probe the aqueous systems and reveal the hidden information. A radar chart named an aquagram was used to visualize and compare the absorbance spectral patterns of waters at different temperatures. For the spectra acquired at a constant temperature of 30 °C, the discrimination analysis of different water samples failed to produce satisfying results. However, under perturbation by increasing the temperature from 35 to 60 °C, the absorbance spectral pattern of different waters displayed in aquagrams revealed different, water-specific dynamics. Moreover, it was found that aged processed water changed with the temperature, whereas the same processed water, when freshly prepared, had hydrogen bonded structures unperturbed by temperature. In summary, the aquaphotomics approach to the NIRS analysis showed that the water absorbance spectral pattern can be used to describe the character and monitor dynamics of each water sample as a complex molecular system, whose behavior under temperature perturbation can reveal even subtle changes, such as aging and the loss of certain qualities during storage.

**Keywords:** aquaphotomics; water; discrimination; temperature; near-infrared spectroscopy (NIRS); perturbation

## 1. Introduction

Water is an essential element of life. The availability of water resources has influenced the human settlements, development of civilizations and public health throughout history. Drinking or potable water comes from different natural sources, such as rivers, lakes, springs and groundwater resources, and various water purification systems were developed and used for the processing and production of water suitable for the human diet. Given the extreme importance that water has for the health and quality of human life, there is a constant need for the development of new methods of control and water quality monitoring. Some parameters, such as the ion content, concentration, ratio of certain ions and pH, among others, have been previously proposed to serve as indicators of water

quality. These chemical and physical parameters of water have been measured by a variety of methods, such as gas chromatography (GC), high-performance liquid chromatography (HPLC), nuclear magnetic resonance (NMR) and many others. However, these measurement systems usually require high installation costs and often sophisticated skills for the preparation of the samples. Furthermore, the currently employed methods may simply be inadequate to capture all of the information relevant for the thorough description of water as a complex system. The prevailing approach toward the characterization of water and the definition of its quality is focused on what solutes are present in it and in what quantities, while completely disregarding that water, together with all of the present solutes, represents a dynamical system, and, as such, is more than a sum of its parts. Therefore, there is a need for better, more sensitive methods for the exploration of water and for the definition of new parameters that are able to effectively describe water systems. These new methods should enable the discrimination and comparison of different water types based on their dynamics and functionalities, in addition to the existing parameters that describe their chemical content and physical and biological properties.

The vibrational spectroscopy: Raman, mid-infrared (IR) and near-infrared (NIR) are widely accepted as useful methods for water quality evaluation because of their sensitivity and potential for real-time quality monitoring [1]. NIR spectroscopy, being non-destructive, fast, requiring minimal or no sample preparation and allowing longer path lengths due to the relatively weak water absorbance, offers excellent possibilities for this purpose. Furthermore, the first overtone of the OH stretching vibration, commonly known as the first overtone of water, located between 1300 nm and 1600 nm possesses many water absorbance bands arising from specific water molecular structures that are shown to have different functionalities by a novel science method called aquaphotomics [2]. The effect of temperature on spectral changes in water in this region is a well-researched topic confirmed by principal component analysis (PCA), two dimensional correlation spectroscopy and other types of analyses [3–5]. Moreover, the perturbation of samples using temperature can serve as an excellent tool for obtaining additional information about the water molecular structure and how it is related to a certain functionality, which is an object of study [6–9].

The aim of aquaphotomics is to provide a complete understanding of water as a multi-element dynamic system that can be described by its multi-dimensional spectra consisting of different absorption bands related to both the overtones and the combination of the stretching and bending vibration of OH [2]. Since different solutes in different concentrations result in different water structures, aquaphotomics offers the new possibility of using the water spectrum as a mirror in which alterations in all present solutes, as well as other external influences, would be reflected. This approach has so far been successfully applied for various purposes, from simple chemical and physical measurements of water solutions to water and food quality monitoring and disease diagnostics [6]. For example, Gowen et al., 2015 successfully measured concentrations of different salts using an aquaphotomics approach (with an estimated limit of detection of 1000 ppm), thus proving that, by measuring the changes in water spectra, caused by any solute, since monoatomic salts do not even absorb in near-infrared, the concentrations of solutes can be measured [10]. Kovacs et al., 2016 reported changes in the water structure of ground water observed by NIR spectra and aquaphotomics, which occur as a consequence of the alterations in the solute content [11]. The authors concluded that the water absorbance spectral pattern could indeed serve as a measure of the qualitative change in the water molecular arrangement and a holistic marker for water quality monitoring. This is further used by Muncan et al., 2020 to propose the utilization of the water spectral pattern as an indicator of water quality for water produced by a purification system intended for household use [12].

Perturbation spectroscopy, on the other hand, was shown to be a very useful approach in studying systems with only subtle differences. This approach refers to the process of acquiring spectra under various perturbations, such as consecutive illuminations, a changing temperature or dilution, in order to induce spectral changes in the respective systems [7,13,14]. This way, by studying how respective systems react to perturbation,

subtle differences between systems can be revealed. This approach was used to reveal the different functionalities of prion protein alloforms [13], and has recently been intensely used for probing the functionality of different biomolecules and increasing the accuracy of measurements [9,15,16].

In this paper, we present the case where the application of the usual aquaphotomics approach, as used by Kovacs et al., 2016 [11], or Muncan et al., 2014, 2020 [12,17], did not provide a clear discrimination of water samples due to their high similarity. In order to overcome this problem, we employed a temperature perturbation protocol. We propose here the application of NIR spectroscopy and aquaphotomics for the exploration of water structural changes under temperature perturbation in order to characterize different waters and discriminate between natural, processed and aged samples of drinking waters.

Therefore, the objective of our research is to introduce a new temperature perturbation protocol for revealing hidden information in the spectra of water and to present this information by water absorbance spectral patterns that reflect different dynamics of different waters and, as such, can be used as an integrative marker for the comparison and discrimination of waters, as well as for water quality monitoring.

## 2. Materials and Methods

### 2.1. Water Samples

Nine different water types from different geographical regions and processed water were used in this study: pure water (Q), tap water samples from Tianjin city (Tap) and mineral waters from Changbai (Cha), Ganten (Gan), Kunlun (Kun) and Tibet (Tib). Processing of waters was performed using procedure with multi-media filtration device (Jilin tasly mineral spring beverage Co., Ltd., Jilin, China). The processing was applied to produce the processed waters from the samples of pure water (PQ) and Changbai mineral water (Cha2014 and Cha2016). The processed samples of Changbai water (Cha) were produced on two occasions: in 2014 (Cha2014) and 2016 (Cha2016). All processed waters (PQ, Changbai2014 and 2016) have a pH up until 24 h after the multi-media filtration (activated carbon filter) was applied.

### 2.2. Methods

Physico-chemical parameters of analyzed waters were measured using conventional methods. Ion content ($Na^+$, $K^+$, $Mg^{2+}$, $Ca^{2+}$ mg/L) was assessed by the reference method in China (GB/T 8538-2008) at the SGS-CSTC standards technical services Co., Ltd., except for Gan, Tib and Kun water samples, for which we used information provided on the labels from the bottles (Table 1). All $^{17}$O-NMR experiments were conducted on a superconductor spectrometer (Bruker, Advance 400 MHz, Bremen, Germany). $^{17}$O spectra were obtained at 54.2 MHz and 298 K. The uniform field was adjusted by standard method with 90% $H_2O$ + 10% $D_2O$. All of the measurements were performed once using one sample replicate (with the similar lot number for each mineral water). The pH was determined as an average of six repeated measurements by the universal method using the pH meter (Mettler Toledo, OH, USA). All $^{17}$O-NMR experiments were conducted on a superconductor spectrometer (Bruker, Advance 400 MHz, Bremen, Germany).

NIR transmittance spectra were acquired using a FT-NIR spectroscopy (Bruker, MPA, resolution: 8 cm$^{-1}$, 64 scans) using 1 mm path length sample cell. Spectra were acquired in the region from 1100 nm to 2400 nm at 0, 6 and 24 h after opening the bottles on the different days for 3 repeated measurements. Temperature perturbation was achieved by varying the temperature from 30 °C to 60 °C in 5 °C increments and the water temperature values in the cuvette were measured by using an infrared thermometer. The total number of spectra was 378 (7 different temperatures × 3 different times after opening the bottles × 2 sample replicates × 3 repeated measurements × 3 consecutive scans) for each water.

**Table 1.** Physico-chemical parameters of analyzed water samples.

| Type of Water | Na$^+$ (mg/L) | K$^+$ (mg/L) | Mg$^{2+}$ (mg/L) | Ca$^{2+}$ (mg/L) | pH at 25 °C (±SD) | $^{17}$O-NMR (MHz) |
|---|---|---|---|---|---|---|
| Pure water (Q) | 0 | 0 | 0 | 0 | 6.86 (0.16) | 76.7 |
| Processed Q (PQ) | 6.3 | 0.3 | 0.0 | 0.3 | 9.81 (0.18) | 53.2 |
| Changbai (Cha) | 8.5 | 4.2 | 9.4 | 12.5 | 7.82 (0.10) | 96.5 |
| Processed Changbai (Cha2014) | 45.0 | 1.8 | 7.4 | 0.2 | 8.33 (0.02) | 82.5 |
| Processed Changbai (Cha2016) | 37.7 | 0.8 | 4.8 | 2.0 | 8.72 (0.04) | 72.0 |
| Ganten * (Gan) | 8.0 | 5.25 | 5.05 | 8.5 | 6.29 (0.08) | 140.6 |
| Tibet * (Tib) | 47.5 | 9.0 | 10.5 | 65.0 | 8.24 (0.04) | 65.6 |
| Kunlun * (Kun) | 22.5 | 2.0 | 45.0 | 45.0 | 8.19 (0.04) | 120.1 |
| Tap water (Tap) | 29.6 | 5.0 | 21.7 | 34.5 | 8.00 (0.06) | 90.8 |

* The source of information was bottle labels.

### 2.3. Multivariate Analysis

All multivariate analyses were carried out by software Pirouette ver. 4.5 (Infometrix, WA, USA). Several types of analyses were performed with different objectives and following the protocol of aquaphotomics data analysis [14] to ensure robustness and consistency.

Hierarchical cluster analysis (HCA) was performed with the goal to examine natural groupings of water samples (clusters) according to the physico-chemical parameters measured by reference methods first, and then, second, using the NIR spectra. The clustering is based on their similarity, which is expressed by the Euclidian distance between clusters in the multidimensional space of variables chosen to describe the samples. The value of distance equal to 1 corresponds to the identical samples, whereas the value of distance equal to 0 corresponds to the most different samples.

Before multivariate spectral analysis, the spectral preprocessing was applied in the form of standard normal variate transformation (SNV) [18] to correct the baseline effects, which are commonly present in the NIR spectra in order to reveal finer spectral features.

To explore the magnitude of spectral changes in waters caused by the temperature perturbation, spectral subtraction was performed. The subtracted spectrum was chosen to reveal the differences in terms of pH, processing, aging and ion content.

The principal component analysis (PCA) [19] was employed in order to explore the spectral data and look into the major sources of variation.

In order to further explore differences in spectra of waters under temperature perturbation, discrimination analysis—soft independent modelling of class analogies (SIMCA) [20,21]—was performed on the SNV transformed and smoothed (Savitzky-Golay, 25 pts) [22] spectra of waters, from which average of pure water (Q) at the corresponding temperatures was subtracted. The resulting interclass distance was analyzed as a function of temperature. This approach revealed different sensitivity and stability of waters to temperature change. The same analysis of interclass distance was used to explore differences between aged and freshly produced waters and processed and unprocessed waters. The objective of this analysis was to further explore differences between unprocessed and processed waters and whether processed water changes over time (2 years).

In order to evaluate which water absorbance bands are most affected by the change in temperature, the SIMCA analysis was performed on nine separate SNV transformed spectral datasets—one for each water under temperature perturbation. The discriminating powers for the thus developed classification models revealed the variables—i.e., wavelengths with highest ability to discriminate between classes of the same water at different temperatures.

From the most influential variables that were found in the respective SIMCA models, 12 wavelengths were selected for the visualization of the absorbance spectral pattern of each water on aquagrams [14,23]. The following 12 wavelengths were selected: 1344, 1364, 1372, 1382, 1398, 1410, 1438, 1444, 1464, 1474, 1492 and 1518 nm. The aquagram type used for this purpose was "classic" [14], where displayed absorbance is normalized and averaged for different sample groups, and displayed on radial axes defined by the 12 selected water absorbance bands.

$$\text{The normalized absorbance is calculated as}: \ A'_\lambda = \frac{A_\lambda - \mu_\lambda}{\sigma_\lambda},$$

where $A_\lambda{'}$ is a normalized absorbance displayed on the aquagram, $A_\lambda$ is the absorbance after SNV transformation, $\mu_\lambda$ is the mean of all spectra for the examined group of samples after transformation, $\sigma_\lambda$ is the standard deviation of all spectra for the examined group of samples after transformation and $\lambda$ are the selected wavelengths. In order to standardize comparison to some known sample, subtraction of average spectrum of pure water at all temperatures was performed and is represented by a zero line on the aquagram.

## 3. Results

### 3.1. HCA Discrimination of Waters Based on Conventionally Used Physico-Chemical Parameters (Ions Type and Concentration, pH and $^{17}$O-NMR)

The nine different water types are listed in Table 1, with their general information including the concentration of ions ($Na^+$, $K^+$, $Mg^{2+}$, $Ca^{2+}$), pH and $^{17}$O-NMR values. The results of HCA are represented by dendrograms in Figure 1, where the dendrogram in the upper panel (Figure 1a) shows natural groupings of water samples described only using information on their ion content, whereas, in the lower panel (Figure 1b), in addition to the ion content, the pH and $^{17}$O–NMR values were included as descriptors. When the analysis was performed using only concentrations of ions ($Na^+$, $K^+$, $Mg^{2+}$, $Ca^{2+}$), the HCA algorithm found six clusters of similar water samples for the similarity level of 0.8 (Figure 1a). In this case, pure (Q) and processed pure water (PQ) show a high degree of similarity (blue cluster), and so do Changbai (Cha) waters, which are processed in different years—in 2016 (Cha2016) and 2014 (Cha2014) (orange cluster). Moreover, if waters were assessed based only on their ion content, waters that are sourced from different geographical locations— Changbai (Cha) and Ganten (Gan)—could not be well separated (magenta cluster). If additional parameters, such as the pH and $^{17}$O–NMR, are included in the analysis, for the level of similarity of 0.8, the waters are classified into eight groups (Figure 1b), with waters Cha2016 and Cha2014 in one cluster, meaning that physico-chemical parameters are not enough to successfully discriminate between waters processed in different years, i.e., conventional parameters are not able to provide information on the age of the water.

**Ion content**

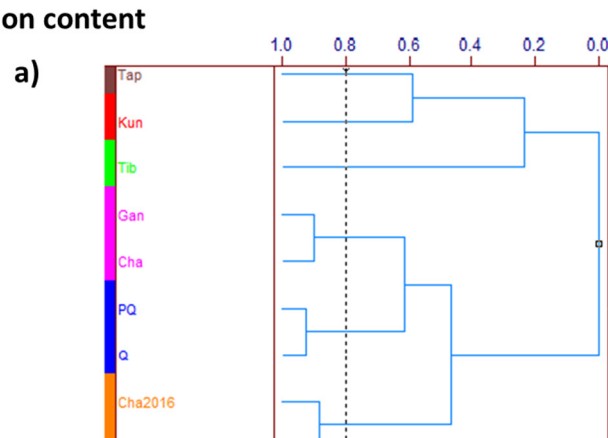

**Ion content, pH, and $^{17}$O –NMR**

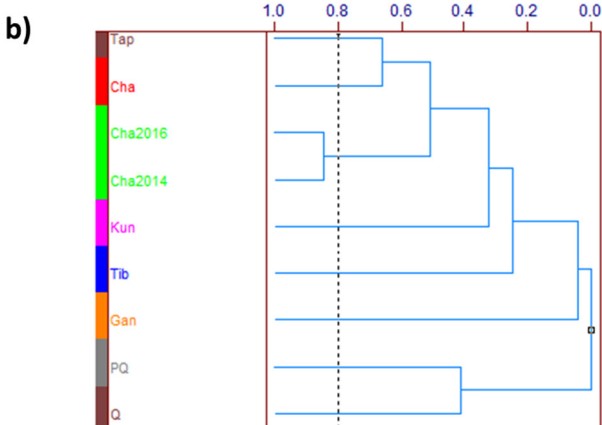

**Figure 1.** Dendrograms of HCA analysis performed using auto-scaled physico-chemical parameters (Table 1) (Panel **a**,**b**). The nine different waters were classified into 6 groups based on ion type (panel **a**) or 8 groups when pH and $^{17}$O-NMR values were used in addition to the ion content (panel **b**). The similarity between the thus found clusters was 0.8.

### 3.2. HCA Discrimination Based on the NIR Spectra of Waters under Temperature Perturbation

An experiment was conducted to discriminate between the various types of waters using NIR spectra in the region of the first overtone of water (1300–1600 nm). NIR absorbance spectra of different waters at temperatures varying from 30 to 60 °C (Figure S1), show that the peak of the first overtone spectra experiences a shift towards lower wavelengths ("blue shift") [3,24] from 1450 to 1428 nm with an increasing temperature, with an isosbestic point around 1442 nm (magnified view in Figure S1), which is consistent with the well-known influence of temperature on water spectra [5,25].

The results of the HCA analysis performed on the SNV transformed spectra of waters at a constant temperature (30 °C) are presented in Figure 2. At the level of similarity of 0.8, four groups of waters can be observed in the dendrogram. The clustering of waters based on the NIR spectra is different compared to the clustering based on the conventional index (Figure 1a,b), which suggests that the NIR spectra of waters are taking into account more than just the ion content. However, at a constant temperature, it was still not possible to successfully discriminate between the waters, and thus the temperature protocol is proposed to perturb each water and enhance subtle differences between spectra.

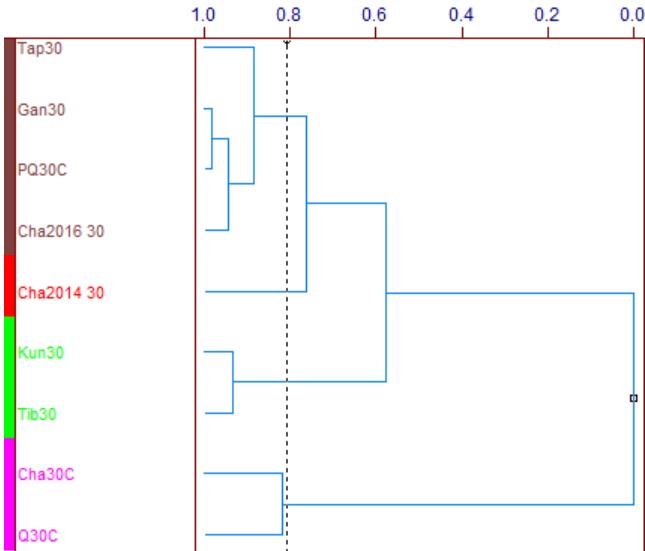

**Figure 2.** Dendrograms of HCA analysis performed using SNV transformed NIR spectra of waters acquired at 30 °C. The nine different waters were classified into 4 groups based on the similarities in the spectral characteristics. The similarity between thus found clusters was 0.8.

### 3.3. Difference Spectra

The difference spectra presented in Figure 3 were calculated between the averaged spectra at all temperatures for the groups of waters defined as: (1) aged water (produced in 2014, Cha2014) and the freshly produced water (produced in 2016, Cha2016); (2) low-ion content waters (Gan, Cha, PQ, Q, Cha2016, Cha2014) and high-ion content waters (Tap, Kun, Tibet), where this grouping was defined based on the results of the HCA analysis, with a similarity level < 0.226 (Figure 1a); (3) low pH waters (pH < 7.5) and high pH waters (pH ≥ 7.5); and (4) processed (PQ, Cha2014, Cha2016) and unprocessed waters (Q, Cha) (Figure 3, the colors are green, red, black and blue for the difference spectra 1, 2, 3 and 4, respectively). All difference spectra show two main peaks at around 1409–12 nm and in the region of 1487–92 nm. In terms of hydrogen bonding, these two peaks correspond to water structures with weak and strong hydrogen bonds, respectively [2,3]. If difference spectra are understood as representing the average difference between water groups caused by the increasing temperature, then the highest difference was observed between the aged and freshly produced water, then the low-ion content and high-ion content waters, the low pH and high pH waters and, lastly, the smallest difference was observed between the processed and unprocessed water.

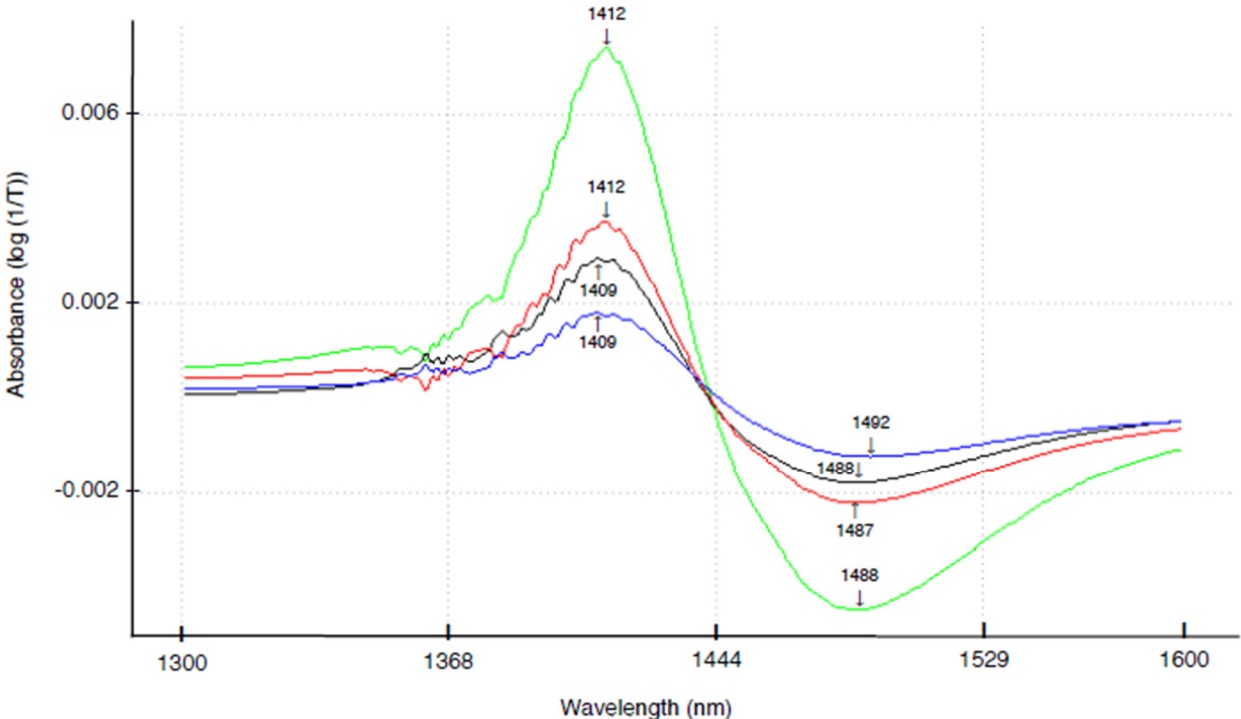

**Figure 3.** Difference spectra indicate the regions of largest spectral difference during temperature perturbation between: Aged water (Cha2014) and freshly produced water (Cha2016)—green line, low ion content waters (Q, PQ, Cha, Cha2014/2016, and Gan) and high mineral content waters (Tib, Kun, and Tap)—red line, low pH waters ($p < 7.5$) and high pH waters ($\geq 7.5$)—black line, and processed water (PQ, Ch2014 and Cha2016) and non-processed water (Q and Cha)—blue line.

### 3.4. PCA Analysis

The PCA scores plots of the first three principal components reveal that the major source of variation in the spectral data is the changing temperature (Figure S2). The second factor was most strongly related to the temperature. The explained variance for temperature effects indicates the percentage of 0.488477% in factor 2. The scores plot factor 2–factor 4 still shows some degree of separation between classes of different waters along the factor 4 axis (Figure 4a). The loading vector of the fourth principal component (PC) shows the specific peaks corresponding to assigned water structures with weak and strong hydrogen bonds in the first overtone region (Figure 4b) [2,3]. However, the clear separation of waters based on the geographical origin, processing or age using PCA was not feasible.

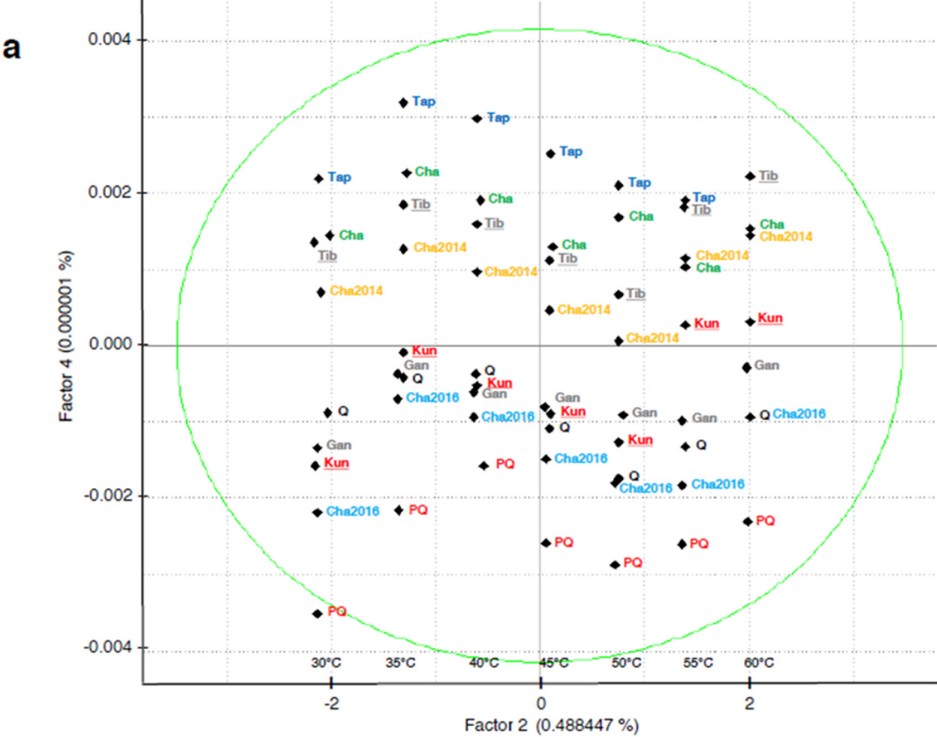

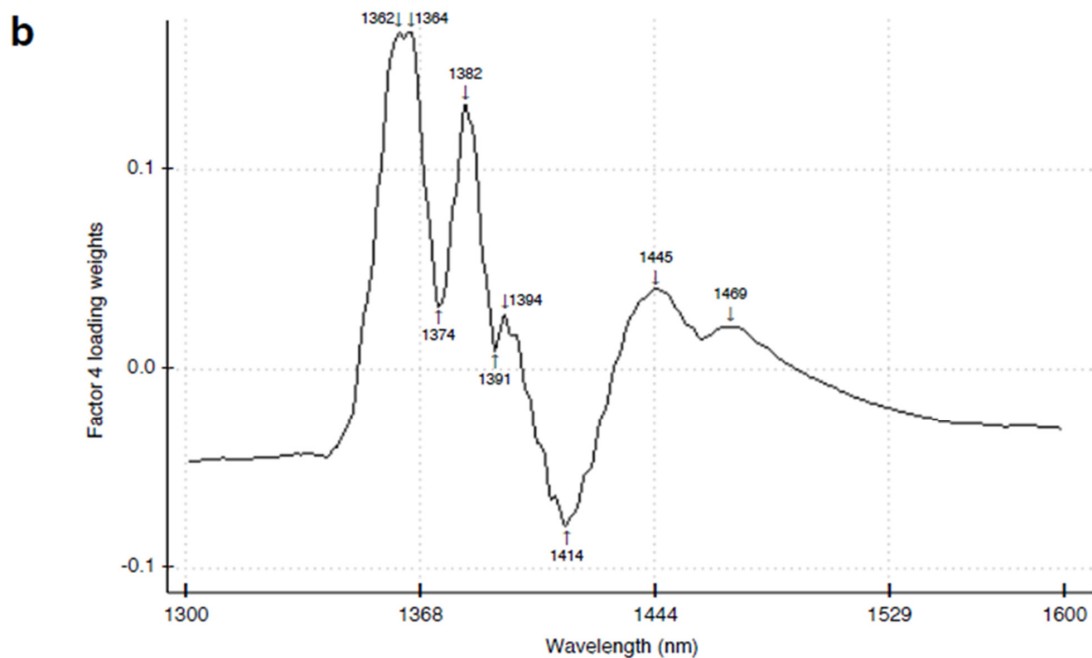

**Figure 4.** The score and loading plot of the principal component analysis for nine different waters. (**a**) The score plot of factor 2 and 4 (Tap: blue, Cha: green, Tib: gray with an underline, Cha2014: yellow, Q: black, Gan: gray, Kun: red with an underline, Cha2016: light blue, and PQ: red). The numbers in parentheses next to the factors 2 and 4, indicate the percentages of the explained total variance. (**b**) The loading of factor 4 shows the importance of certain wavelengths in the respective spectral pattern of this principal component. All indicated wavelengths are water absorbance bands in the first overtone of water.

### 3.5. Interclass Distance Study Reveals Different Temperature Dependence of NIR Spectra of Water

The interclass distances of eight different waters after the pure water (Q) average spectrum was subtracted that were analyzed as a function of temperature (Figure 5) showed the existence of two groups of waters (Figure 5a). One group of waters (dotted lines with triangular marks, Cha: green, Cha2014: yellow, Tib: gray and Kun: red) showed large changes in interclass distances with an increasing temperature i.e., a large sensitivity to temperature perturbation. The second group of waters (solid lines with circular marks, PQ: red, Cha2016: light blue, Gan: gray and Tap: dark blue) showed comparably small changes in interclass distances when the temperature was increased i.e., a relative insensitivity to temperature perturbation compared to the first group. The percentage of correctly classified mineral waters was dependent on the level of temperature perturbation and, on average, was 88% (from minimum 67% to maximum 100% (Table S1).

The SIMCA distance between the classes of unprocessed pure water (Q) and processed pure water (PQ) shows minor changes in the range of 0.89 to 0.78 with an increasing temperature (Figure 5b). On the other hand, the distance between classes of aged water (Cha2014) and freshly produced water (Cha2016) increased up to 2.01 with the temperature increase (Figure 5c). These results indicate that the increase in temperature leads to growing differences in the water molecular matrix, and, considering the comparison using the difference spectrum of aged and freshly produced water (Figure 3, spectrum shown in green), we can conclude that aged water is more sensitive to temperature perturbation than the freshly processed water.

### 3.6. Aquaphotomics Unraveled Hidden Information in Water Spectra under Temperature Perturbation

The objective of this analysis was to further explore differences between unprocessed (Cha, Q) and processed waters (Cha2014, Cha2016 and PQ) and whether processed water changes over time (2 years). In order to evaluate which water absorbance bands are mostly affected by the change in temperature, the SIMCA analysis was performed on nine separate SNV transformed spectral datasets—one for each water under temperature perturbation. The discriminating powers for the thus developed classification models revealed the variables—i.e., the wavelengths with the highest ability to discriminate between classes of the same water at different temperatures (Figure S3). The common peaks were found at wavelengths that can be assigned to specific water molecular species: at around 1407 nm for free water, at 1444 nm for the water cluster with dimer (S1) and 1517 nm for the stretching and bending vibration of water v1 (symmetric stretching vibration of OH) and $\nu 2$ (bending vibration of OH) i.e., strongly bound water [2]. Regarding the regions indicated with grey bars in Figure S3, from the most influential variables that were found in the respective SIMCA models, 12 wavelengths were selected for the visualization of the absorbance spectral pattern of each water on aquagrams.

Aquagrams presented in Figures 6 and S4 display the averaged and normalized absorbance values at selected wavelengths after the SNV transformation and subtraction of the average spectrum of pure water at all temperatures (Q—zero line in the center of the aquagrams). All of the wavelengths presented as radial axes on the aquagrams belong to the WAMACs (water matrix coordinates $C_i$, i = 1–12) ranges, where specific water molecular conformations absorb [2]: $H_2O$ asymmetric stretching vibration, $\nu 3$ (C1: 1344 nm), water hydration shell OH-$(H_2O)_{1,2,4}$ (C2: 1364 nm), symmetrical stretching fundamental vibration and $H_2O$ asymmetric stretching vibration, $\nu 1 + \nu 3$ (C3: 1372 nm), water hydration shell, OH-$(H_2O)_{1,4}$ and/or superoxide, $O_2$-$(H_2O)_4$ (C4: 1382 nm), free water and free OH-, [S0] (C5: 1398 nm, C6: 1410 nm), water molecules with different numbers of hydrogen bonds: water dimer—S1 (C7: 1438 nm (protonated water monomer, hydronium) and C8: 1444 nm, water dimer), water trimer—S2 (C9: 1464 nm), water tetramer—S3 (C10: 1474 nm), highly hydrogen bonded "ice-like" water—S4 (C11: 1492 nm), symmetrical stretching fundamental vibration and doubly degenerate bending of bonded water, $\nu 1$, $\nu 2$ (C12: 1506–1516 nm) [2,26,27].

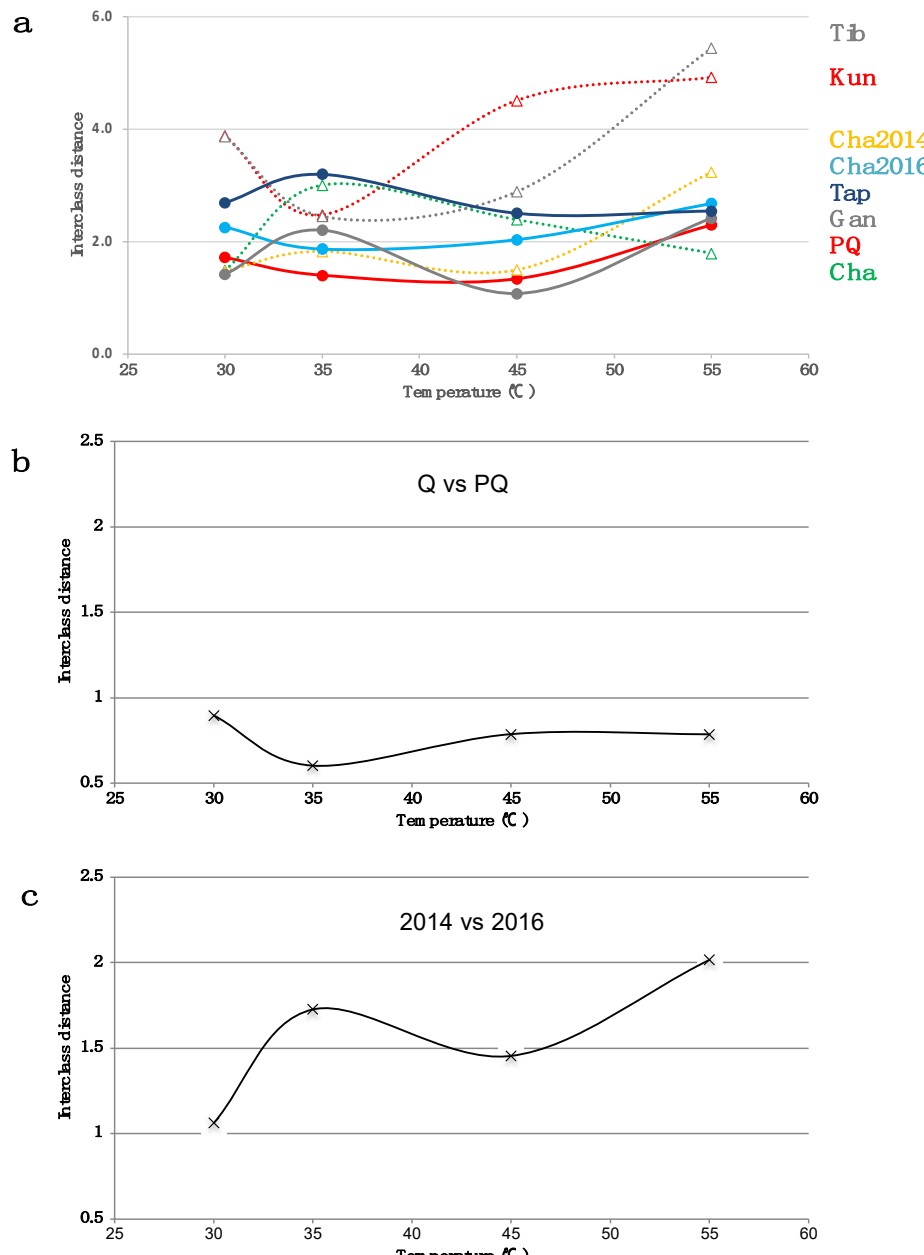

**Figure 5.** SIMCA interclass distance plotted as a function of temperature unravel information in the natural, processed and aged water. (**a**) The interclass distance of the eight different waters compared to pure water (Q). Solid lines with circular marks, PQ: red, Cha2016: light blue, Gan: gray and Tap: dark blue and dotted lines with triangular marks, Cha: green, Cha2014: yellow, Tib: gray and Kun: red. (**b**) The interclass distance between processed water and non-processed water (Q vs. PQ) as a function of temperature. (**c**) The interclass distance between aged and freshly produced processed waters (Cha2014 vs. 2016) as a function of temperature. Spectral data are transformed using the SNV, average Q at the given temperature is subtracted and, finally, the spectra are smoothed using Savitzky Golay filter (25 pts).

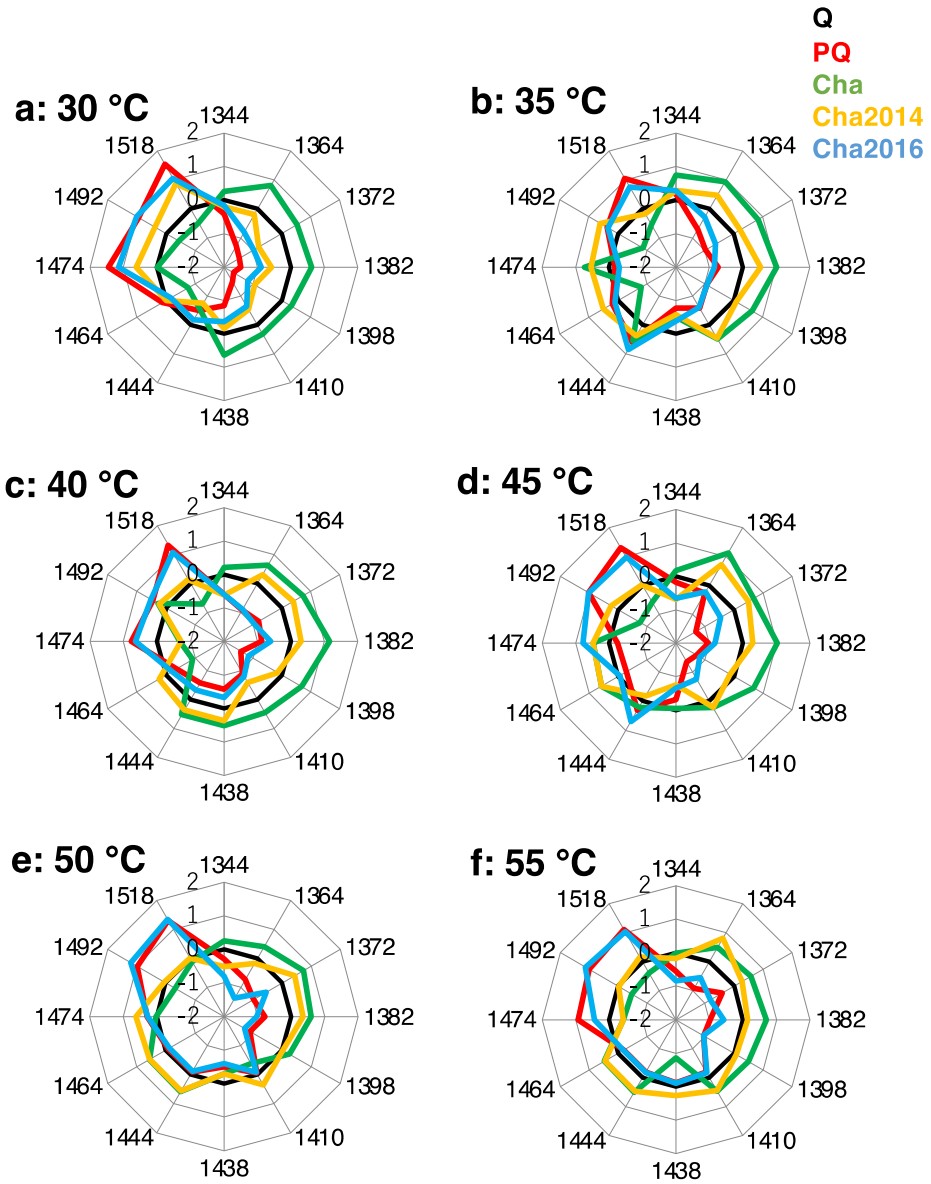

**Figure 6.** The aquagrams displaying the water absorbance spectral pattern of processed and unprocessed waters under temperature perturbation: ((**a**) 30 °C; (**b**) 35 °C; (**c**) 40 °C; (**d**) 45 °C; (**e**) 50 °C; and (**f**) 55 °C). Normalized absorbance presented at radial axes is calculated as follows: an average of pure water at all temperatures was subtracted from the averaged spectra of each water at a specific temperature and SNV transformed. Different color of the lines represents different waters (Cha: green, Cha2014: yellow, Cha2016: light blue and PQ: red). Since all waters are being compared to the pure water, central zero line corresponds to the pure water (Q: black). Spectral data are SNV transformed and average Q across all temperatures is subtracted.

At the beginning of the experiment, under no perturbation by temperature (at a constant 30 °C), the waters show distinctive groupings in the aquagram (Figure 6a). All of the processed waters show a high absorbance in the left part of the aquagram, in the region from 1474 nm to 1518 nm, whereas the unprocessed water Cha shows a high absorbance on the opposite side. The largest differences between the group of processed waters and unprocessed waters can be seen in the region of 1364–1438 nm, which corresponds to free and less hydrogen bonded water, and the region of 1474–1518 nm, which corresponds to more hydrogen bonded water. Based on this observation, we can conclude that the processing of water results in the creation of a more hydrogen bonded water matrix.

Under perturbation, by increasing the temperature (Figure 6b–f), the processed waters show different dynamics. The aged water, which was processed in 2014 (Cha2014), shows the increased absorbance of less hydrogen bonded and free water structures with a temperature increase. The aquagram of this water is shifted towards the less hydrogen bonded water region 1364–1410 nm. On the other hand, despite the increase in temperature, hydrogen bonded water structures remained stable in the freshly processed waters (PQ and Cha2016). These two waters showed almost no changes in absorbance at the region of 1492–1518 nm and at the 1398 nm water band. Interestingly, the aged water Cha2014 with an increasing temperature displays a water spectral pattern that becomes more similar to the unprocessed water (Cha).

In summary, aquaphotomics unraveled the subtleties of the molecular structure in each water by utilizing the temperature perturbation and showing that it is possible to discriminate between natural, processed and aged waters based on their near-infrared spectra when applying aquaphotomics. It further revealed that hydrogen bonded structures remain stable in processed waters (PQ and Cha2016) despite the increase in temperature of up to 60 °C; however, this stability of water structures under temperature perturbation in processed water (Cha2014) is lost two years after the production.

## 4. Discussion

The quality of drinking water is usually assessed by physico-chemical, biological and organic analysis. When the certain standardized criteria in their chemical and microbiological content are met, water is considered safe for drinking. The characterization of waters following this concept is therefore focused on the determination of the cation and anion content, their ratios and physical parameters, such as pH, conductivity, $^{17}$O-NMR and others [11,17]. As many studies have shown so far, the characterization of water by physico-chemical indices often cannot provide a successful discrimination of different waters or give the exact location of their geographical source, let alone determine when the water is produced.

In this regard, our attempt at the discrimination of nine different waters based solely on their ion content is not different. When HCA analysis was performed by using only chemical content information, we could not discriminate between the same water produced in 2016 and 2014, between waters coming from two entirely different geographical locations or if the water was processed using an activated carbon filter or not (Figure 1a). The results were slightly better when two more descriptors were included in the analysis—the pH and $^{17}$O-NMR—but again we could not discriminate between waters produced in different years (2014 and 2016) (Figure 1b). This indicates that attempts in characterizing water solely using its ion content are lacking in information. One of the reasons for this fact is that the ion content does not change with temperature or other possible external influences, and only a few chosen parameters cannot provide a complete evaluation of water as a dynamic molecular system. Interestingly, when the additional parameters of pH and $^{17}$O-NMR are added to the analysis, the results improved. pH as a measure of the concentration of protons (H$^+$) in the solution, which in our case is water, is a property that arises from the formation of a water system and is a result of the interaction of its subsystems—solutes and water together. In addition, $^{17}$O, which is extremely sensitive to hydrogen bonding, together with pH, adds more information about the state of the water system as a whole. It becomes obvious that water as a complex system is more than a simple sum of its constituents, and attempts at characterizing the system by describing the properties of its constituents will always be deficient and inadequate in describing the state of the waters; therefore, the future methods of water characterization should aim to measure parameters that describe the water system as a whole.

NIR spectroscopy has long been accepted as a useful nondestructive and chemical-free probe for the analysis of the water molecular structure. It was previously reported that, in the range of the first overtone of the OH stretching vibration (1300–1600 nm), a number of characteristic prominent bands exist that are related to the specific water molecular

conformations (such as free water molecules, water dimers, trimers, tetramers, strongly bound water, etc.) with different functionalities [2]. The structure of water, which is based on hydrogen bonds, is very sensitive to any internal or external perturbation; hence, whatever change the water experiences, it will have a direct influence on the absorbance bands in this region. By measuring absorbance at specific water absorbance bands (called WAMACs in aquaphotomics—water matrix coordinates), insight into the whole molecular structure of water is gained, and the resulting water spectral pattern (WASP) can be used to describe the state of water as a system. The utilization of this region for the discrimination of mineral waters has already been reported and the water spectral pattern as a holistic marker for water quality monitoring has also been proposed [6,11,12,17].

In the present research, NIR spectroscopy was applied to characterize nine different waters for which the classical approach for discrimination by using physico-chemical parameters failed to produce results. Special emphasis in our research was given to the problem of discrimination between processed and unprocessed waters, and the same water produced in different years, for which classical methods of discrimination proved to be inadequate. To be able to carry this out, we developed a temperature perturbation protocol, during which we performed a spectral acquisition from water systems under an increasing temperature. The changes in the spectra of waters as a reaction to perturbation were then used to characterize their behavior.

The temperature dependence of the NIR spectra of water led in past studies to the proposition of two or three-component mixture models of water [5]. The two-component model was proposed as a result of the observation of a linear relationship, with temperature for the first principal component and the turning point at 38 °C on the second principal component in the principal component analysis of temperature-dependent water spectra [3]. In another study on the influence of temperature (in the range 28–45 °C), the turning point appeared around 37 °C on the third principal component, no clear isosbestic point was observed and, based on the results of principal component analysis (PCA) and multivariate curve resolution-alternating least square (MCR-ALS) analysis, authors concluded that the three-component model would be more appropriate to describe the water system behavior under the influence of temperature [5]. In the present study, NIR spectral plots reveal a changed slope trend at around 35 °C (Figure S1), which is in agreement with both above studies, but, on closer inspection, we could not observe the true isosbestic point, but rather a blurred area around 1441–1443 nm, which is consistent with the results of Gowen et al., 2013 [5]. The change in the trend of water spectra was also noticed in the interclass distance dependence of temperature, where an apparent change occurs somewhere around 35 °C (Figure 5b). Taken together, we concluded that the three-component mixture model could be used to approximate the dynamic behavior of the water systems we analyzed, where the three components would be strongly hydrogen bonded water, weakly hydrogen bonded water and a third—an intermediate form of water [5].

By expressing the SIMCA interclass distance between each of the waters and pure water as a function of temperature (Figure 5a), we were able to observe how each water system reacts to change when the heat is supplied. It can be argued that, in general, the values of the interclass distance are very low to claim a successful discrimination and existence of a difference between the waters, but, in the case of water discrimination, where we start with only subtle differences, maybe this is not the appropriate measurement at all and the rate of changes in the interclass distance should instead be more appropriate because it reflects different dynamics. In this framework, with an absolutely different range of changes in the interclass distance, we can conclude that, with regard to temperature perturbation, processed water shows an insensitivity to temperature (Figure 5a) compared to the aged processed water, which shows a sensitivity to temperature changes (Figure 5b). In other words, by tracking how dynamical water systems evolve under exerted perturbation, if we find a difference in behavior, then it means we found a difference in initial conditions, and, no matter how small and subtle those differences were in the beginning, they determine different future states, different behavior and different functionality. Introducing a water

spectral pattern as a representation of the water structure visualized by aquagrams, we were able to observe and compare the states of our water systems during different stages of perturbation. This enabled further insight into what the differences are between processed and unprocessed water (Cha vs. PQ, Cha2016) and freshly produced and the aged water (Cha2016 vs. Cha2014) (Figure 5).

The aquagrams of processed waters (PQ, Cha2014, Cha2016) show three common features in the beginning of the experiment at 30 °C: a high absorbance at 1492 nm and 1518 nm and a low absorbance at 1398 nm compared to pure water. These three features remain stable throughout perturbation for processed waters PQ and Cha2016; however, this ability to keep these structures stable is lost in the aged water (Cha2014). The high absorbance at 1492 nm would suggest a high number of water molecules making four hydrogen bonds, whereas the high absorbance at 1518 nm would suggest a high amount of bonded water. The location at 1395–1403 nm accounts for the so-called trapped water (TW), which is associated with water molecules trapped in the region of overlapping first hydration shells of ions [26].

We cannot say what the possible consequences of the existence of stable water structures in the processed water could be, because studies on the functionality of different water structures are still very scarce, hence further in vivo clinical studies would be needed to assess the functionality, bioavailability or other properties of this water, which may have health implications. However, since water activity is deeply related to the water structure [28], we can safely say that a different structure of water will lead to different water activity.

In this research paper, we showed that, with temperature perturbation, initially subtle differences between waters can be enhanced, and that, with the water spectral pattern, we can both describe the state of each water in terms of the present water structures and determine whether processing or aging changed the water structure. With this developed protocol and aquaphotomics approach, we defined a new method for the characterization of a variety of so called "structured waters", which are very popular at the market and are both produced by exposing waters to different treatments (such as light, magnetic field, quartz crystals, etc., to name just a few) and supposedly have health effects. Our approach to the characterization of water could then, if coupled with in vivo studies, relate specific water molecular structures or combinations of structures to the final functionality of water and the effects it produces in living systems.

It may be argued that inducing temperature perturbation does not hold promise for the development of rapid monitoring systems and a freshness evaluation; however, a similar effect can be induced by perturbing water with light [13,14], which, in practical applications, would mean to just acquire more consecutive spectra.

## 5. Conclusions

In conclusion, what our study showed is that the temperature perturbation led to different behaviors of the water dynamical systems, the properties of which could be captured, described and compared using a water absorbance spectral pattern.

**Supplementary Materials:** The following are available online at https://www.mdpi.com/article/10.3390/app11199337/s1, Figure S1: The SNV transformed NIR absorbance spectra of nine different waters., Figure S2: PCA of nine different waters under temperature perturbation., Figure S3: The discriminating powers of SIMCA., Figure S4: The aquagrams displaying the water absorbance spectral pattern., Table S1: The percentage of the accurate number exactly classified to each mineral water.

**Author Contributions:** Conceptualization, Y.K., M.Y. and J.-Y.H.; methodology, Y.K., J.M. and D.K.; software, D.K.; validation, Y.K. and J.M.; formal analysis, Y.K.; investigation, Y.K.; resources, J.-Y.H.; data curation, Y.K. and J.M.; writing—original draft preparation, Y.K.; writing—review and editing, J.M., R.T., J.-Y.F. and J.-Y.H.; visualization, Y.K..; supervision, R.T. and M.Y.; project administration, J.-Y.H.; funding acquisition, J.-Y.H. All authors have read and agreed to the published version of the manuscript.

**Funding:** This work was supported by the State Key Laboratory of Core Technology in Innovative Chinese Medicine [20170034]. J.M. gratefully acknowledges financial support provided by Japanese Society for Promotion of Science (P17406). The funders had no role in the study, data collection and analysis, decision to publish or preparation of the manuscript.

**Institutional Review Board Statement:** Not applicable.

**Informed Consent Statement:** Not applicable.

**Data Availability Statement:** Not applicable.

**Acknowledgments:** The authors would like to thank Polachi Navaneetha Krishnan, Zhi-Wen Li, Yan Xu and Yi-Chan Mei for insightful comments and discussion, and Simon Li, Xiao-Hui Ma, Shui-Ping Zhou at Tasly Academy and Kai Sun at Peking University for technical support. J. Muncan gratefully acknowledges the financial support by JSPS Postdoctoral Fellowship for Research in Japan (P17406). This report will be presented as a research finding of Research Project Keio 2040 in the Longevity Initiative at Keio University Global Research Institute (KGRI).

**Conflicts of Interest:** The funders had no role in the design of the study; in the collection, analyses or interpretation of data; in the writing of the manuscript, or in the decision to publish the results.

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
