# Peer review of "Aquaphotomics Reveals Subtle Differences between Natural Mineral, Processed and Aged Water Using Temperature Perturbation Near-Infrared Spectroscopy"

_applsci, doi:10.3390/app11199337_

Round 1

Reviewer 1 Report

The paper entitled “Aquaphotomics reveals subtle differences between natural mineral, processed and aged water using 3 temperature-perturbation near infrared spectroscopy” investigates an interesting and valuable work. The findings are interesting, and the work is valuable. However, I believe that some corrections are needed to improve the manuscript.

Line 54: “ion content, concentration, the ratio of certain ions and pH” -> comma needed before the and word if you use a series of three or more words. (line 73 the same issue)

Line 56: „gas chromatography (GS)” isn’t it GC?

line 92 For example Gowen et al. <- year is missing the same issue in line 95, 100,

line 134 comma not needed after „Kun”

Authors should be consequent in usage of units after the numbers. In some cases it presents without a space, while in some case with space. e.g.: line 141 (region from 1100nm to 2400 nm)

Line 141: „on 3 different days” how long was the difference between the measurement days? Were they following or not?

2.3… Much more details needed for the description of the data analysis. This part should take place here, not in the results.

line: 155-156 – it's a methodology description, which does not take place in the results normally. Please, put these part in the methods/statistics part. And give more information regarding the HCA- for which data was it applied, moreover what type of distance was calculated and so on.

line 175-176. It is methods again.

line 181-183 spectral pretreatment should be detailed in the methods, not in the results.

line 192-194 it is  also rather methods

line 208-209 it is  also rather methods

line 220—223, 235-236 248-252 methods.

Moreover, what type of aquagram was calculated?

Table 1: Did the authors measure (in the case of samples that were analyzed) more repetitive regarding the Ionic content? If yes, why did you not include the standard deviation? The same question related to pH. In order to give more trustable data, the physicochemical measurements should be performed in more repeats also.

Figure 1. Where is panel c? (line 185)

In general the paper revealed interesting results that are useful, however the method part should be more concretized, in special regard to the statistical analysis. It should be included in the methods, not in the results itself in a detailed way.

Reviewer 2 Report

The study aims to apply NIR spectroscopy and aquaphotomics for the exploration of water structural changes under temperature perturbation in order to characterize different waters and discriminate between natural, processed and aged samples of drinking waters.

Aquaphotomics in a growing field that has shown good potential for applications in diverse fields. In addition, the study was detailed and well described in a repeatable manner. Discussions were also well done.

My only concern is the novelty of this study. Some of the authors have already published similar studies: http://www.doiserbia.nb.rs/img/doi/0367-598X/2014/0367-598X1300049M.pdf . The exact difference between their previously published papers and this study needs to be clarified.

In addition, authors concluded that “the temperature perturbation led to different behaviors of the water dynamical systems, properties of which could be captured, described and compared using water absorbance spectral pattern”. This is not new in the field of perturbation spectroscopy and aquaphotomics. Some of the authors even published a paper about the temperature dependence analysis of NIR of liquid water: https://www.sciencedirect.com/science/article/abs/pii/S0167732219331617?via%3Dihub

In summary, I would advise that the novelty of this current study be elucidated in a clear, concise and striking manner.

Round 2

Reviewer 2 Report

The novelty of the study has been clarified now in the manuscript. The overall quality of the manuscript has also, improved and from the purview, it can be accepted for publication.

Thank you.